# What Affects Emotional Well-Being during Travel? Identifying the Factors by Maximal Information Coefficient

**DOI:** 10.3390/ijerph19074326

**Published:** 2022-04-04

**Authors:** Yongfeng Ma, Shuyan Chen, Aemal J. Khattak, Zheng Cao, Muhammad Zubair, Xue Han, Xiaojian Hu

**Affiliations:** 1Jiangsu Key Laboratory of Urban ITS, School of Transportation, Southeast University, Nanjing 211189, China; mayf@seu.edu.cn (Y.M.); 220162605@seu.edu.cn (Z.C.); 233217035@seu.edu.cn (M.Z.); huxiaojian@seu.edu.cn (X.H.); 2Jiangsu Province Collaborative Innovation Center of Modem Urban Traffic Technologies, Southeast University, Nanjing 211189, China; 3Nebraska Transportation Center, 330E Whittier Research Center, University of Nebraska-Lincoln, 2200 Vine Street, Lincoln, NE 68583-0851, USA; khattak@unl.edu; 4Civil Engineering College, Wanjiang University of Technology, No. 333 Huolishan Avenue, Ma’anshan 243031, China; hanxue_echo@163.com

**Keywords:** emotional well-being, maximal information coefficient, feature selection, travel purpose, travel duration

## Abstract

Traveler emotional well-being as a specific domain of subjective well-being has attracted attention across the field of transportation. Studies on identifying factors of travel-related emotional well-being can help policy makers to formulate concrete strategies to improve travelers’ experiences and public health. This research used the Maximal Information Coefficient (MIC) to select important factors which have much influence on emotional well-being during travel. American Time Use Survey data collected in 2010, 2012, and 2013 were used in this study and 10 factors have been selected to illustrate the relationship with emotional well-being, including rest, weekly earnings, activity time for well-being, health, self-evaluation of activities, pain medication taken yesterday, travel purpose, travel duration, weekly working hours and age based on MIC values in Descending sort. Among these 10 selected features, 2 factors, travel purpose and travel duration, are related to travel contexts; the other factors are related to personal and social characteristics. It is found that an individual’s physical condition and self-evaluation of activities have much influence on travel-related emotional well-being, while traveling mode and interaction during travel have a relatively small impact on emotional well-being compared to other identified factors. This finding is different from previous research findings. The paper presents traffic strategies related to improving emotional well-being of travelers while traveling based on the findings from this research.

## 1. Introduction

Well-being has been suggested as another indicator of social development level besides income and economic prosperity. The United Nations General Assembly adopted resolution 65/309 Happiness, “Towards a Holistic Definition of Development” in July 2011, and the United Nation’s Sustainable Solutions Network has published the World Happiness Report every year since 2012. This trend is consistent with the growing evidence that happier people tend to be heathier and successful across work, marriage, sociality, and so on [1,2,3,4,5,6].

Travel plays an important role in people’s daily life, and travel experiences influence their emotional well-being towards travel as well as life. Evidence shows that travel-related factors affect subjective well-being, transportation planners, engineers, and policymakers can better understand the inner connections between traffic systems and EWB, and thus can formulate transportation policies to improve the travel happiness and life happiness of the public [7]. Hence, analyzing the influence of travel factors on travel happiness and EWB has received great interest from researchers in the field of well-being.

SWB consists of three components including positive affect, negative affect, and cognitive SWB [8]. The balance of positive affect and negative affect during people’s daily activities is defined as emotional well-being (EWB) [9]. Subjective well-being (SWB) is defined as the degree to which individuals positively evaluate the overall qualities of their lives according to their living criteria [10]. As Friman et al. [11] judgment that EWB is obtained by the frequencies, average intensity, or different duration of positive and negative affect [12,13,14].

Past research has explored the relationship between travel factors (e.g., mode, duration, congestion) and EWB [15,16,17]. In these elaborate studies, whether and how specific travel modes and corresponding travel duration contribute to travel-related EWB were discussed thoroughly. Recently, Chen et al. [18] proposed a new process by combining optimal scale regression, factor analysis, and analytic hierarchy to isolate different factors, and based on these factors having greater impacts on travel-related EWB, they provided some suggestions on planning, engineering as well as policy to improve travel-related EWB.

Although many methodologies have been proposed, they have not yet adequately solved the many challenges associated with factors selection for travel-related EWB, in particular, the importance or influence of different factors is difficult to compare, and it is still an open problem that needs addressing. In this study, we made use of the maximal information coefficient (MIC) to identify the factors’ importance to travel-related EWB. Compared to one similar previous work [18], this paper which explored maximal information coefficient to extract the factors contains the following new contributions; (a) using MIC to extract factors which have advantages of equitability [19], and are better to identify a linear or nonlinear relationship between variables than a conventional statistic model; (b) evaluating the different factors’ relevance and importance to EWB directly by MIC, while the models based on utility theory focus on some relationship between travel-related EWB and certain factors such as travel mode and travel duration; (c) we illustrate self-evaluation of activities has a significant impact on travel-related EWB, a factor that was ignored in our previous work; (d) our results demonstrate that travel mode and interact during travel have a relatively small impact on emotional well-being than travel purpose and travel duration, which is different from previous research [18,20,21] and partly consistent with [22,23]. 

The remainder of this paper is organized as follows: Section 2 presents a review of previous studies investigating the relationship between travel factors and EWB followed by Section 3 introducing feature selection, mutual information, and MIC. The ensuing Section 4 describes American Time Use Survey (ATUS) data. Section 5 illustrates the case study based on ATUS data. Section 6 including suggested policies, pros, and cons of using MIC to extract factors completes this paper.

## 2. Literature Review

Bergstad et al. [24] confirmed that the effect on affective and cognitive SWB of satisfaction with daily travel is both direct and indirect via satisfaction with the performance of activities. Many studies have investigated the relationship between travel mode and EWB. Cao et al. [20] used a structural equation model to explore how light rail transit affects people’s satisfaction with life. Chng et al. [21] investigated how commute mode and neighborhood public transport connectivity were associated with EWB. De et al. [15] provided an overview of studies concerning travel and well-being and suggested that travel behavior affects well-being through experiences during (destination-oriented) travel, activity participation enabled by travel, activities during (destination-oriented) travel, trips where travel is the activity, and through potential travel (or motility).

Much research has indicated that many factors have a relationship with SWB. SWB is influenced by socioeconomic and demographic factors. Socioeconomic factors involve income, education, work, unemployment welfare, public insurance, and so on. Demographic factors may include age, gender, health, ethnicity, and attitudes [25]. Travel-related emotional well-being is special domain of well-being, also required to consider the impacts of the above factors besides travel factors. The following section mainly reviews the relationship between travel factors and emotional well-being.

Ettema et al. [10] questioned whether and how changes in travel context may cause changes in SWB. Furthermore, Friman et al. [26] investigated whether satisfaction with daily travel is related to life satisfaction and EWB, and found that satisfaction with the commute is related to travel mode and travel time. Friman et al. [11] addressed the question of how the commute to work changes positive versus negative and active versus passive mood experienced after completion of the commute. Morris et al. [22] studied the relationship between traveler’s mood and travel mode and concluded that mood is no worse during travel than on average and travel only has a small total impact on mood.

Additionally, some researchers have illustrated that traffic congestion influences EWB to some extent. Morris et al. [17] examined how drivers experience their emotions during their driving in peak periods and concluded that congestion has a limited impact on drivers’ moods. Olsson et al. [27] found that work commute influenced the balance of positive and negative affect.

Travel-related EWB as a specific domain of SWB has attracted relatively little attention and needs further investigation. The affecting factors framework has been constructed including personal characteristics, social development, economics, built environment, travel environment, and so on [10,11,25]. Although published research presents some interesting insights into EWB, the studies are limited and not sufficient to illustrate how changes in travel factors may influence EWB; for example, trip purpose as an important factor has not attracted much attention. The majority of available research is focused on the relationship between travel mode, travel duration, and EWB. Travel duration is considered to another important factor influencing the EWB, especially during the daily commute. Stutzer et al. [28] found that people with longer commute time report systematically lower SWB. Wheatley et al. [29] studied how travel time and travel mode influence EWB. 

Zhu et al. [30] examined how the mode, duration, purpose, and companionship characteristics of a trip shape the positive and negative emotions. They found that travel for discretionary purposes is generally associated with higher levels of positive emotions and lower levels of negative emotions than travel for work or household maintenance. 

In this paper, a feature selection method based on maximal information coefficient is utilized to select factors which have an influence on travel-related EWB. ATUS data collected by the American Bureau of Labor Statistics in 2010, 2012, and 2013 are used in this study. Avoiding the complex data preprocessing and model fitting, MIC can directly evaluate each factor’s information gain to EWB. Based on ranked MIC results, factors with greater values were selected as they have much more influence on travel-related EWB than those with smaller values.

## 3. Method Description

Generally speaking, data collected may contain some redundant or irrelevant features, and can thus be removed without incurring much loss of information. Feature selection is such a process that selects a subset of relevant features from available features. Using feature selection, we can simplify models by using features as few as possible; thus making the models easier to interpret, reducing the models’ training times, avoiding dimensionality curse, and enhancing the generalization of models.

Assume that there is a data set: $D=\left(F,C\right)=\left\{f_{1},f_{2},K,f_{m},c_{1},c_{2},K,c_{k}\right\}$; here, $F=\left\{f_{1},f_{2},K,f_{m}\right\}$ is sample features (also called factors or variables), and $C=\left\{c_{1},c_{2},K,c_{k}\right\}$ is sample categories. The goal of features selection is to identify some features from *F* which have strong relationships with *C*. A feature selection algorithm is composed of two parts: one is a search process to select new feature subsets, and the other is an evaluation function to score the different feature subsets. Suppose $J\left(f_{i}\right)$ is the value of the evaluation function for feature $f_{i}\left(1=1,K,m\right)$, it reflects the importance of this feature, $J\left(f_{i}\right)\in \left[0,1\right]$ where 0 corresponds to having nothing with *C*, and 1 means the most important feature. The bigger the value is, the more important the feature is. That means $f_{i}\left(i=1,2,\ldots ,m\right)$ contains more information with greater $J\left(f_{i}\right)$; therefore, it can be chosen as an important factor.

### 3.1. Mutual Information

Mutual information is often used to obtain evaluation functions for feature selection. Assume that $I\left(X,Y\right)$ is the mutual information between variable *X* and *Y*, which means when the distribution of *X* is determined, *X* contain $I\left(X,Y\right)$ information to determine the distribution of *Y*. Assume $p\left(x\right)$ and $p\left(y\right)$ represent the marginal probability of *X* and *Y* separately, $p\left(x,y\right)$ is the joint probability function of X and Y, the formulas of mutual information are presented in Equations (1) and (2) [31].
(1)$$I\left(X,Y\right)=\int_{y}^{.}\int_{x}^{.}p\left(x,y\right)\log_{.}\frac{p\left(x,y\right)}{p\left(x\right)p\left(y\right)}dxdy$$
(2)$$I\left(X,Y\right)=H\left(X\right)+H\left(Y\right)-H\left(X,Y\right)$$
where $H\left(X\right)$ is information entropy of *X*, calculated by the formula $H\left(X\right)=-\int_{x}^{.}p\left(x\right)\log p\left(x\right)dx$ similarly, $H\left(Y\right)=-\int_{y}^{.}p\left(y\right)\log p\left(y\right)dy$ and $H\left(X,Y\right)$ is the joint entropy of *X* and *Y*, calculated by the formula $H\left(X,Y\right)=-\sum_{x}\underset{y}{\sum }p\left(x,y\right)\log p\left(x,y\right)$.

Although mutual information evaluates the dependence between two variables, mutual information is not a true metric in the mathematical sense because of failing to obey the triangle inequality. With continuous variables, mutual information is sensitive to the discrete progress of continuous variables. Thus, it is not suitable to apply mutual information as an evaluation function to select features directly. Thus, the maximal information coefficient is chosen to be the evaluation function in order to overcome the problems of mutual information.

### 3.2. Maximal Information Coefficient

On the basis of mutual information, MIC evaluates mutual information of two variables. If a relationship exists between two variables, then a grid can be drawn on the scatterplot that partitions the data to encapsulate that relationship. Still assuming *X* and *Y* are a pair of variables, then according to [19]:If any functional relationship exists between *X* and *Y* at any interval, then $\mathrm{mic}\left(X,Y\right)$ is closer to one;Only two variables are dependent on each other, $\mathrm{mic}\left(X,Y\right)$ is equal to zero;If $\mathrm{mic}\left(X,Y\right)$ tends to be larger, the variable *X* and *Y* are more relevant. MIC of variable *X* and *Y* can be formulated as follows,
(3)$$\mathrm{mic}\left(X,Y\right)=\max_{x*y<B}\frac{I\left(X,Y\right)}{\log_{2}\min\left(X,Y\right)}$$
where $I\left(X,Y\right)$ is mutual information between variable *X* and *Y*; B is function of sample size, which is usually set as $B=n^{0.6}$ and n is sample size.

According to MIC characteristics, if two variables have greater MIC, they are more relevant and the determined variable contains more information to determine the distribution of the other variable. Thus, we utilized MIC as the evaluation function for feature selection in this study. We calculated the MIC of each variable and EWB, then ranked the variables according to the MIC values, thus fulfilled factors selection of EWB.

## 4. Data Description

The American Bureau of Labor Statistics has conducted a series of national investigations on how, where, and with whom Americans spend their time in each of more than 400 detailed activities since 2003 [32]. The respondents are interviewed about how they spent their time from 4 a.m. on the pre-selected day to 4 a.m. on the interviewed day. The ATUS questionnaires are detailed and all the massive amount of data is recorded in the ATUS database. More information on the collection of this data in ATUS is available at https://www.bls.gov/tus/ (accessed on 20 June 2017).

In 2010, 2012, and 2013, the ATUS questionnaire included a well-being module as a special supplement to the ATUS database. ATUS questionnaire data along with registered data on income, residence, work, and activities, etc., enabled the inclusion of sociodemographic indicators, health indicators, and travel characteristics along with emotional well-being. This well-being module asked respondents to report how they felt during different activities and described their feelings as happiness, fatigue, stress, sadness, and pain. The measurement scale for each feeling ranges from 0 to 6, where 0 refers to the least and 6 means the most; taking happiness as an example, 0 means not happy at all, and 6 means very happy. According to previous research by [33], who estimated EWB by the Affect Balance Scale method, we merged the five emotions into a value by subtracting the mean of unpleasant emotions from the mean of positive emotions. The merged values range from −6 to 6 and are skewed distribution. Thus, the value is normalized into the interval [0, 1] with log-transformed and then isometric discrete into five numbers $y_{i}\in \left[0,\;1,\;2,\;3,\;4\right]$ to represent different levels of EWB; higher numbers representing higher EWB [17].

This three-year ATUS database contains the respondent’s ID, socioeconomic and demographic characteristics, and activity information. ATUS well-being data includes respondent’s ID, activity information, and feelings during activities. According to the respondents’ ID and activity information, the travel-related records are selected during the process of matching ATUS data records. Moreover, we removed some records with missing values that note respondent’s failure to answer specific questions. After data preprocessing, the matched dataset for this study contains 16568 records obtained from the 2010, 2012, and 2013 ATUS datasets. Each record in the matched data comes with various socioeconomic, demographic, and travel information on the respondents. Table 1 presents description of selected variables of this study.

## 5. Experiments and Results

In this study, we first calculated MIC values between variables and EWB. Afterward, normalized MIC value were calculated by using the method of min–max normalization. Next, ranked variables were determined by the normalized MIC value of each variable in descending order. The variables with greater MIC were selected as factors with more important to travel-related EWB.

### 5.1. Results of MIC

The MIC values between each variable under consideration and travel-related EWB are calculated according to formula 1 and 3, and all the variables’ MIC values have been normalized to [0, 1]. Then, all variables are ranked by their MIC values in descending order. Figure 1 shows a plot of the ranked results. If we set a threshold of 0.14, we can extract 10 features with MIC significantly greater than the rest variables. These are rest, weekly earnings, activity time for well-being, health, meaning, pain medicine, travel purpose, travel duration, weekly working hours, and age. The remaining variables have relatively small MIC values, implying they almost have no impact on travel-related EWB.

### 5.2. Results Analysis

The aforementioned results on the ATUS data produced some interesting findings. From the perspective of qualitative analysis, most of the chosen features reflect some subjective characteristics. Rest, health and pain medication history reveal the respondent’s physical conditions. It is inferred that a good physical condition means positive attitudes to conduct activities. Additionally, self-evaluation of activities also has significant impact on travel-related EWB, revealing that respondent’s cognition affects their travel-related EWB. Weekly earnings and weekly working hour reflect socioeconomic characteristics affecting travel-related EWB to a certain extent.

Among the selected factors, only travel purpose and travel duration are related to travel contexts. Travel purpose and travel duration have significantly higher MIC values, which is consistent with previous findings. Travel purpose affects the traveler’s cognition to travel-related EWB until the end of trip. Consistent with previous research [16,28], travel duration is shown to be strongly associated with travel-related EWB. Although previous research [20,21] has investigated the effects of travel mode on travel-related EWB, our research shows that travel mode contains a relatively little information to determine the distribution of travel-related EWB value, which implies that travel mode has relatively small impact on travel-related EWB compared to the other selected features.

## 6. Transport Policy Suggestions Based on Our Results

Transport policy has a significant impact on personal well-being by providing mobility and accessibility. It will help to formulate more reasonable transportation policies by exploring the factors that affect travel happiness, and help to improve the public travel experience, happiness, and health. The result shows that the most important indicators which affect travel-related EWB are as follows: rest, weekly earning, health, and activity time for WB. Therefore, the following suggestions are made.

### 6.1. Guarantee Good Rest and Health

Good rest and health have a significant impact on improving travel-related EWB. Good rest requires a good living environment; furthermore, living environment is closely related to the selection of residential land and the planning and design specifications of the residential area. Unfortunately, the “Urban” Residential Area Planning and Design Standards” (GB50180-2018) of China does not have any corresponding clauses on how to reduce the impact of urban road noise on residential areas. Therefore, it is recommended to build green spaces and reduce the impact of urban road noise on residential areas.

In addition, it is more cost-effective to invest in the construction of rest areas on the expressway to increase the travel-related EWB of individuals. Strengthening management to release travelers from experiencing travel fatigue is also an effective strategy to improve travel happiness. At the same time, urban environmental quality has a significant impact on the health of residents. Since the main sources of urban environmental pollution are traffic noise and automobile exhausts, this can be reduced by using electric environmentally protective vehicles and green fuel substitutes. Regular inspection and maintenance of vehicles can directly reduce emissions pollution as well.

What is more, urban transport structure can be optimized with the guidance of comprehensive public transport, which can effectively alleviate the exhaust of urban road vehicles’ pollution. The measures are as follows: improvement of urban public transport infrastructure; renewal and transformation of public transport vehicles; rational planning of urban public transport lines and density; setting bus lanes on roads with certain conditions; and improvement of urban public transport coverage and share rate. Taking a series of traffic-calming measures in streets and residential areas, such as adjustment of road network structure, community entrance design, parking planning, construction of slow traffic facilities, planting arbors, etc., can effectively reduce traffic noise pollution and provide environmental protection for resident′s good rest and health.

### 6.2. Encourage Traveler Interaction during Travel

Interaction during travel plays an important role in improving travel-related EWB. Therefore, family travel and group travel are encouraged by adopting more favorable ticketing and charging models, setting up facilities such as seats in buses, subways, and stations that facilitate group exchanges. In tourist attractions and places of entertainment facilities with a large number of tourists, multiparticipant projects can be set up. Moreover, consideration can be given to strengthening the interaction between travelers and transport facilities, such as building up electronic bus-boards and installing multimedia in vehicles to reduce pressure sensation during travel. Constructing a 15 min living circle residential area will improve people’s accessibility to various living services and facilities, which will help encourage people to travel with relatives and friends and strengthen interactions.

### 6.3. Improve the Travel Environment

Travel characteristics affect residents’ travel-related EWB to a large extent, including travel purpose, travel duration, and travel mode. Studies have shown that there is a significant connection between travel purpose and travel-related EWB. Therefore, it is particularly important to establish a transportation environment that is conducive to more autonomous travel. At the same time, shortening travel time is beneficial to improve travel-related EWB. Some studies have shown that people who travel longer are more likely to have negative emotions such as fatigue, stress, and sadness [23]. In order to shorten travel duration, clear road grades and functions, clear road rights, and optimized traffic structure should be made in the planning stage. Additionally, P + R facilities and internal public transport facilities should be improved to ensure the good accessibility of urban traffic.

Travel modes also have an impact on travel-related EWB. Studies have shown that nonmotorized travel modes such as walking have the highest levels of well-being. Urban slow transit systems consist of a nonmotorized vehicle system and pedestrian system. This is the main mode of short-distance travel, and is also an indispensable transportation mode for connecting medium- and long-distance travel with public transportation.

In the inner part of the city, the nonmotorized lane layout should be rationally designed in combination with the actual traffic flow and section status of the road as well as in combination with the construction of the landscape pedestrian belt. Recreational bicycle lanes along rivers should be planned to create a pleasant nonmotorized traffic environment together with the landscaped pedestrian corridor. In order to construct a pleasant pedestrian system, commercial pedestrian streets, assembly squares, and pedestrian crossings should be beautified.

## 7. Conclusions

Numerous studies have illustrated that different dimensions impact travel-related EWB including socioeconomic, demographic, and travel-related factors. Our study is an extension of such work. The research reported herein focused on identifying the factors which have influence on travel-related EWB in a comprehensive manner, as well as the importance of each factor contributing to travel-related EWB, rather than on the relationship between travel-related EWB and some certain factors such as travel mode and travel duration. To illustrate the connections between this large number of factors and EWB, this research utilized maximal information coefficient to analyzes the ATUS well-being dataset. A total of 10 factors were selected by using MIC based on mutual information from the constructed initial feature set.

To avoid complex data preprocessing for traditional statistical models, this research explored an easier and more-efficient method to select influential factors for travel-related EWB by calculating and ranking the MIC of each features. Maximal information coefficient directly measures the degree of correlation between emotional well-being and factors. Higher MIC value means that the factor contains more information to determine the distribution of emotional well-being value. Thus, the selected features have greater influence on EWB, and are better candidates in the future quantitative analysis. If we set a lower threshold, we could choose more factors, which affords us the flexibility to investigate the influence of further factors on travel-related EWB.

This paper extracted important factors influencing travel-related EWB based on the maximal information coefficient. Ongoing research includes some in-depth quantitative analysis on how and why these selected factors influence travel-related EWB. For example, how to trip purpose increase or decrease travel-related EWB. In a word, relationships between the identified features and EWB are worthy of being further explored in future research.

## Figures and Tables

**Figure 1 ijerph-19-04326-f001:**
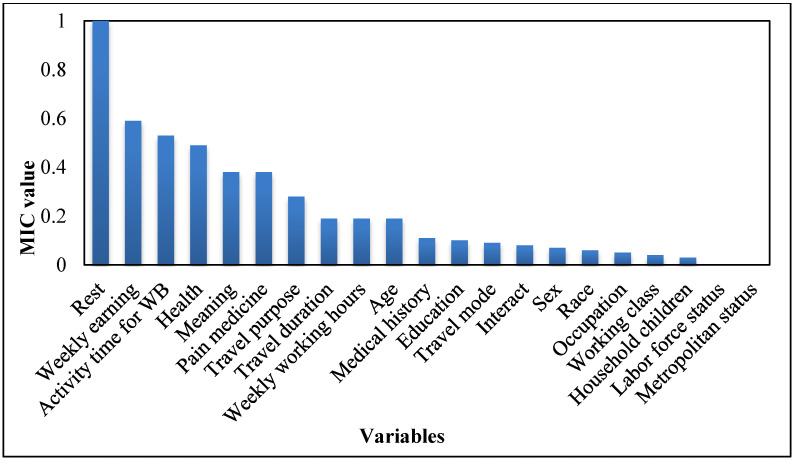
Rank of feature’s maximal information coefficient.

**Table 1 ijerph-19-04326-t001:** Description of socioeconomic, demographic, and travel characteristics.

Factors	Variable Label	Description
Socioeconomic	Occupation	Categorical variable, 6 values to describe major occupation category
	Weekly earning	Continuous variable, weekly earnings at main job
	Labor force status	Categorical variable, 5 values to describe labor force status (employed—at work, absent; unemployed—on layoff, looking; not in labor force)
	Work class	Categorical variable, 8 values to describe individual class of worker
	Weekly working hours	Continuous variable, total hours usually worked per week
	Metropolitan status	Categorical variable, metropolitan status (2000 definitions)
	Activity time for WB	Continuous variable, total time spent in all activities for well-being module
Demographic	Age	Continuous variable, age, ranged from 15 to 85
	Sex	Categorical variable, gender, female or male
	Education	Categorical variable, 16 values to describe the respondents’ education level
	Race	Categorical variable, 21 values to describe different racial types
	Household children	Categorical variable, presence of household children (0—No, 1—Yes)
	Health	Ordinal variable, from 1 to 5, where 1 means excellent and 5 means poor
	Medical history	Categorical variable, any sickness told by doctors before 5 years, 0—No, 1—Yes
	Pain medicine	Categorical variable, taking any pain medication yesterday, 0—No, 1—Yes
	Rest	Ordinal variable, from 1 to 4, where 1 means very rested and 4 means not at all
Travel-related	Travel mode	Categorical variable, where were you during activities (car, truck, walking, bus and subway train etc.)
	Travel duration	Continuous variable, duration of activity in minutes
	Interact	Categorical variable, whether or not interacting with anyone during activities
	Meaning	Categorical variable, from 0 to 6, how meaningful did you consider what you were doing (self-evaluation to activities)
	Travel purpose	Categorical variable, 71 values which are six-digit activity codes (e.g., travel related to personal care)

## Data Availability

All data used during the study were provided by a third party, the American Bureau of Labor Statistics. The data can be download for free at https://www.bls.gov/tus/ (accessed on 20 June 2017).

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
