# Peer review of "What Affects Emotional Well-Being during Travel? Identifying the Factors by Maximal Information Coefficient"

_ijerph, 2022, doi:10.3390/ijerph19074326_

Round 1
Reviewer 1 Report
Dear authors,
Dear editor,
Given the limitations of the study that the authors referred in the manuscrit and in the answers, it seems there is nothing to add in the manuscript.
In this sense, I think the manuscript is ready for publication, with the need of review the bibliography numeration as it is not ordered (authors should check the journal norms).
Best regards.
Author Response
No. ijerph -1651756
Title: What affect emotional well-being during travel? Identify the factorsby Maximal Information Coefficient
Authors: Yongfeng Ma12, Shuyan Chen12,*, Aemal J. Khattak 3, Zheng Cao1, Muhammad Zubair 1, Xue Han4 and Xiaojian Hu1
Authors’ response
Thanks the anonymous reviewers for their valuable suggestions and comments to improve the quality of this paper. The following are the main changes introduced in the paper according to the reviewer comments.
Reviewer #1 Comment
Dear authors, given the limitations of the study that the authors referred in the manuscript and in the answers, it seems there is nothing to add in the manuscript. In this sense, I think the manuscript is ready for publication, with the need of review the bibliography numeration as it is not ordered (authors should check the journal norms).
Thanks the anonymous reviewers for their valuable suggestions and comments to improve the quality of this paper. According to your question and comments I have carefully revised the manuscript and we have checked and improved the journal norms. We have ordered the bibliography numeration.

Reviewer 2 Report
Thank you very much for the opportunity to read the proposal of the article “What affect emotional well-being during travel? Identify the factors by Maximal Information Coefficient”.
The Authors used the methodology of Maximal Information Coefficient (MIC) to select main factors which have much influence on emotional well-being during travel.
This manuscript contains a many errors:
- Line 220: Why is 21 a total of the set variables? Lack of description.
- Line 4: “1”, “2”, “3”, “4” - number written or printed above the line. The authors don't apply the ijerph Microsoft Word template (https://www.mdpi.com/files/word-templates/ijerph-template.dot) or LaTeX template (https://www.mdpi.com/authors/latex) or a Free Format Submission (https://www.mdpi.com/journal/ijerph/instructions#freeformatsub) to manuscript preparation.
- Line 78: citation style. The authors don't apply the ijerph Microsoft Word template (https://www.mdpi.com/files/word-templates/ijerph-template.dot) or LaTeX template (https://www.mdpi.com/authors/latex) or a Free Format Submission (https://www.mdpi.com/journal/ijerph/instructions#freeformatsub) to manuscript preparation.
- Line 123: no space between „on” and „[12, 14, 16].
- 3: “max” – no italic.
- Table 1 - no clear dividing lines factors: socioeconomic, semographic and travel-related.
- Figure 1: The lack of detailed description of the X-axis and Y-axis.

Author Response
No. ijerph -1651756
Title: What affects emotional well-being during travel? Identify the factorsby Maximal Information Coefficient
Authors: Yongfeng Ma12, Shuyan Chen12,*, Aemal J. Khattak 3, Zheng Cao1, Muhammad Zubair 1, Xue Han4 and Xiaojian Hu1
Authors’ response
Thanks the anonymous reviewers for their valuable suggestions and comments to improve the quality of this paper. The following are the main changes introduced in the paper according to the reviewer comments.
Reviewer #2 Comment
Thank you very much for the opportunity to read the proposal of the article “What affects emotional well-being during travel? Identify the factors by Maximal Information Coefficient”.
The Authors used the methodology of Maximal Information Coefficient (MIC) to select main factors which have much influence on emotional well-being during travel.
This manuscript contains a many errors:
Line 220: Why is 21 a total of the set variables? Lack of description.
Thank you very much to pint out about lack of description about the 21 variables because the dataset is too large so we have divided dataset in number of variables although I have made the corrections.
Line 4: “1”, “2”, “3”, “4” - number written or printed above the line. The authors don't apply the ijerph Microsoft Word template
We have improved and carefully revise it according to the ijerph Microsoft Word template.
Line 78: citation style. The authors don't apply the ijerph Microsoft Word template
We have improved and carefully revise citation style according to the ijerph Microsoft Word template.
Line 123: no space between „on” and „[12, 14, 16].
We have made the corrections.
3: “max” – no italic.
We agreed and changed it.
Table 1 - no clear dividing lines factors: socioeconomic,semographic and travel-related.
We have made clear dividing lines factors: socioeconomic, Demographic and travel-related.
Table 1. Description of socioeconomic, demographic and travel characteristics.
|
Factors |
Variable label |
Description |
|
Socioeconomic |
Occupation |
Categorical variable, 6 values to describe major occupation category |
|
Weekly earning |
Continuous variable, weekly earnings at main job |
|
|
Labor force status |
Categorical variable, 5 values to describe labor force status (employed-at work, absent; unemployed-on layoff, looking; not in labor force) |
|
|
Work class |
Categorical variable, 8 values to describe individual class of worker |
|
|
Weekly working hours |
Continuous variable, total hours usually worked per week |
|
|
Metropolitan status |
Categorical variable, metropolitan status (2000 definitions) |
|
|
Activity time for WB |
Continuous variable, total time spent in all activities for well-being module |
|
|
Demographic |
Age |
Continuous variable, age, ranged from 15 to 85 |
|
Sex |
Categorical variable, gender, female or male |
|
|
Education |
Categorical variable, 16 values to describe the respondents’ education level |
|
|
Race |
Categorical variable, 21 values to describe different racial types |
|
|
Household children |
Categorical variable, presence of household children (0-No, 1-Yes) |
|
|
Health |
Ordinal variable, from 1 to 5, where 1 means excellent and 5 means poor |
|
|
Medical history |
Categorical variable, any sickness told by doctors before five years, 0-No, 1-Yes |
|
|
Pain medicine |
Categorical variable, taking any pain medication yesterday, 0-No, 1-Yes |
|
|
Rest |
Ordinal variable, from 1 to 4, where 1 means very rested and 4 means not at all |
|
|
Travel-related |
Travel mode |
Categorical variable, where were you during activities (car, truck, walking, bus and subway train etc.) |
|
Travel duration |
Continuous variable, duration of activity in minutes |
|
|
Interact |
Categorical variable, whether or not interacting with anyone during activities |
|
|
Meaning |
Categorical variable, from 0 to 6, how meaningful did you consider what you were doing (self-evaluation to activities) |
|
|
Travel purpose |
Categorical variable, 71 values which are six-digit activity code (e.g. travel related to personal care) |
Figure 1: The lack of detailed description of the X-axis and Y-axis.
We have explained the detailed description of the X-axis and Y-axis in Figure 1
peer-review-18385615.v1.pdf(/user/review/displayFile/25363070/NcPgwA9d?file=review&report=18385615)
All these changes have been done.

This manuscript is a resubmission of an earlier submission. The following is a list of the peer review reports and author responses from that submission.
Round 1
Reviewer 1 Report
Dear authors,
Dear Editor,
Thank you very much for the opportunity to read the manuscript “What affect emotional well-being during travel? Identify the factors by Maximal Information Coefficient”.
This paper intends to identify the main factors that influence the emotional well-being during travel using the Maximal Information Coefficient with the objective to support the design of the transport policies.
The discussed subject is very interesting within the transport and urban mobility studies and in debate nowadays. However, the paper presents some limitations to be published as it is in the actual moment, as:
- Title should include the reference to the study area and period of time;
- Section 1 – Introduction and Section 2 – Literature review are similar and quite repetitive.
- The introduction do not present a clear objective. Section 1 could be simplified, presenting the main subject, the objectives and the trigger that originated this study and the structure of the paper, sending the discussion about Subjective well-being and the methodologies to measure it to the literature review; or other possible solution it to merge section 1 and 2.
- The literature review could be reinforced creating a summary table with the methodologies, main indicators and/or results obtained.
- The paragraph of lines 153-163 should have the source. Is this methodology in general used in other type of studies?
- The data is quite updated (American Time Use Survey from 2010, 2012 and 2013). Perhaps there is no more recent data, but this is a strong limitation that should be referred;
- It is not clear what is the 16568 records obtain from 2010, 2012 and 201 ATUS dataset. This corresponds to the number of individuals? This was for all USA or for a specific area? Is this a sample? Why to use 3 different years?
- Main results based on data and reflecting the used methodology is totally missing. There are no tables, graphs or any support to justify the text;
- Figure 1 should be the nearest possible from its first reference (line 242) – Figure 1 (the only figure) is missing
- Section 5 is one of the most important of the paper, however this is not discussed in-depth. There is no data to support the text (graphs or tables). Indeed, the result analysis is almost null;
- The transport policies suggestion in Section 6 are based on literature? The references are scarce in this section, and this suggestions are not clearly related with the obtained results
- Section 7 should be focused on the main conclusions. Discussion should be present previously (within the result section or not). The main limitations of this work are not identified here.
- The conclusion section should be reformulated as this includes “discussion”, as discussion of the results should be next to the results and before the policy proposals;
- About formatting, the references in the text are not according to the journal`s formatting rules:
“Authors should consider the format of the references within the text: “In the text, reference numbers should be placed in square brackets [ ], and placed before the punctuation; for example [1], [1–3] or [1,3]. For embedded citations in the text with pagination, use both parentheses and brackets to indicate the reference number and page numbers; for example [5] (p. 10). or [6] (pp. 101–105).”
- This reference is not cited in the text:
Morris, E. A. , Guerra, E. , 2015. Are we there yet? Trip duration and mood during travel. Transp. Res. Part F: Traffic Psychology and Behavior, 412 33, 38-47. 413. https://doi.org/10.1016/j.trf.2015.06.003 414
Despite the relevance of the theme, in the actual form, the paper do not present the high quality to publish in IJERPH.
Best regards
Reviewer 2 Report
A description of the comments to the article is provided in the annex.

Reviewer 3 Report
Dear Authors, Your article is interesting, first of all, not so much the subject of research, as the formulation of the question and the chosen method. The ideas of the article have the potential for development.
At the same time, there are a number of important remarks.
Principle remarks.
- In section 3.1, you refer to Kraskov's et al. results (lines 170...171) - formulas (1) and (2). But in Kraskov's et al. original article, other formulas are given for determining mutual information ?(?,?). In Kraskov's et al. article, these are formulas (1) and (5) and they are presented in a different form.
- Figure 1 illustrating the results of the research is missing. Also, it would be nice to show the exact calculated MIC values for all 21 variables.
- There are contradictions between the information of lines 242 and 246.
- You have analyzed 16568 records obtained from the ATUS 2010, 2012 and 2013 datasets with differentiation on 21 grounds, but you make unambiguous conclusions without taking into account the distribution of data into groups. Meanwhile, it is known that age will significantly influence the preferences of travelers. I didn't find this in the article.
- Section 5.2 - Results Analysis - needs to be radically improved.
- Section 6 - Transport policy - it would be nice to also improve.
Other remarks.
1. References to the sources used are designed without taking into account the requirements of the journal.